# Use of Irrigation Device for Duct Dilatation during Sialendoscopy

**DOI:** 10.3390/ijerph192214830

**Published:** 2022-11-11

**Authors:** Giulio Pagliuca, Veronica Clemenzi, Andrea Stolfa, Salvatore Martellucci, Antonio Greco, Marco de Vincentiis, Andrea Gallo

**Affiliations:** 1Otolaryngology University Unit, Santa Maria Goretti Hospital, 04100 Latina, Italy; 2Department of Sensorial Organs, ENT Section, “Sapienza” University of Rome, 04100 Rome, Italy; 3Department of Oral and Maxillofacial Sciences, “Sapienza” University of Rome, 00161 Rome, Italy

**Keywords:** salivary gland, sialendoscopy, irrigation device, salivary duct

## Abstract

Background: Continuous irrigation of the duct with isotonic saline is one of the fundamental stages of a successful sialendoscopic procedure. It allows for an adequate luminal distension for the removal of debris and mucous plugs and for the conservative treatment of strictures. This procedure, which commonly involves the use of a medical syringe, can be laborious, and it is often necessary to interrupt irrigation during surgery due to the high resistance to saline. Setting: Academic university hospital. Method: We propose the use of an irrigation device which consists of a high-pressure syringe barrel, an ergonomic piston handle, and a gauge used to monitor the inflation and deflation of balloon catheters. The system allows for a simple and safe dilation, ensuring good visualization of the salivary duct lumen during sialendoscopy. Conclusions: The irrigation system described can be widely used to perform a diagnostic or interventional sialendoscopy more effectively than with a typical manual irrigation procedure.

## 1. Introduction

Sialendoscopy, a minimally invasive procedure first mentioned by Kats in 1991, represents a continually expanding field and is becoming the preferred technique for diagnosing and managing salivary gland obstruction [1,2,3]. The continuous irrigation of the duct is one of the cardinal points of an effective sialendoscopic procedure. Irrigation with isotonic saline solution is necessary to overcome the sphincter-like contractile mechanism that keeps the duct in a collapsed state and allows an adequate luminal distension so that intraductal structures and pathologic changes of the duct can be clearly visualized. Irrigation also plays a pivotal therapeutic role, allowing for the removal of debris and mucous plugs from the ductal system and the conservative treatment of strictures, solving alone many of the most common obstructive conditions of the salivary glands [4]. Most surgeons opt for a manual irrigation process using a common medical syringe. This procedure can be laborious, and it is often necessary to interrupt irrigation during surgery due to the high resistance to saline. In this paper, we propose the use of an irrigation device that allows for the performance of a simple and safe dilation, ensuring good visualization of the salivary duct lumen during sialendoscopy.

## 2. Materials and Methods

During minimally invasive balloon dilation procedures, balloon inflation devices are widely used to inflate the balloon with fluid, monitor the pressure in the balloon during the procedure, and deflate the balloon after dilation. In ENT surgery, these devices are used mainly to dilate obstructed sinus ostia during sinuplasty in patients suffering from sinusitis, or to perform a Eustachian tube (ET) dilation to treat obstructive ET dysfunction, or during subglottic stenosis balloon dilation.

In this paper, we describe our experience with a balloon inflation device adapted to perform a constant irrigation of the salivary duct during sialendoscopy. Any patient showing an obstructive or inflammatory disease of the salivary glands could benefit from sialendoscopy performed with this new irrigation system. In our daily surgical practice, we use this device for irrigation on all patients undergoing sialendoscopy, regardless of the cause that generates the symptomatology.

The device (Figure 1) consists of a high-pressure syringe barrel, an ergonomic piston handle, and a gauge used to monitor the inflation and deflation of balloon catheters (Disposable Inflation Device, Cook Medical, AL, USA). In order to perform an adequate and constant dilation of the salivary glands, this device is connected with the irrigation channel of the sialendoscope and to a saline solution bottle through a three-way stop-cock (Figure 2). The stop-cock is positioned to allow the saline solution to flow from the bottle to the syringe of the device, which can be filled with 20 cc of solution (Figure 3a). The procedure is performed using an all-in-one Erlangen-type sialendoscope (Karl Storz, Tuttlingen, Germany) which are 1.1 mm in diameter for diagnostic and 1.3 or 1.6 mm for interventional sialendoscopy. Once the syringe of saline is filled, the device is locked, and the three-way stop-cock is rotated to allow the fluid to pass from the syringe to the sialendoscope (Figure 3b). The grip on the piston handle must be rotated clockwise to push the saline to the sialendoscope in order to increase the pressure in the ductal system until reaching the desired value. A gauge is used to monitor the pressure of saline in the salivary ducts. As a portion of the saline flows through the ductal orifice between the endoscope and the ductal walls and a portion is absorbed by the gland, the piston handle is rotated slowly and steadily to avoid the gradual decrease of the pressure in the ductal system. Turning the grip on the piston handle counterclockwise will decrease the pressure in the ducts. Once the saline solution in the syringe is depleted, the procedure can then be easily repeated, if needed.

## 3. Discussion

Sialendoscopy is a procedure for the diagnosis and treatment of salivary obstructive and inflammatory diseases [5,6,7,8]. It can be performed under local or general anesthesia depending on the level of difficulty of the individual case. Continuous rinsing of the duct during the procedure is necessary to achieve its dilatation and consequently an adequate visualization of the lumen to allow the advancement of the scope into the duct system [9]. Irrigation allows us not only to obtain an effective dilation of the duct during endoscopy but also to treat some obstructive sialadenitis. The opportunity for administering drugs, such as corticosteroid preparations, directly into the ductal system provides an interesting point of access for salivary inflammatory disease therapy in adults and children [10,11,12,13]. Furthermore, the treatment of juvenile recurrent parotitis is based on dilation of the Stensen duct, and it is often performed using strong saline solution irrigation [14]. Continuous washing with a saline solution allows for the removal of mucous plugs from the lumen ducts, while the hydrostatic pressure on the ductal system can be considered a useful method to treat the strictures, especially in the case of multiple stenoses (often observed as a result of radioiodine therapy), when the use of a sialoballoon or of the sialendoscope itself does not allow the dilation of the whole stenotic tract [15]. In these cases, however, it is necessary to achieve and maintain an adequate pressure value during the surgical procedure to obtain a durable dilation and an effective washing of the ducts. Most surgeons opt for a manual irrigation performed by an assistant using a typical medical syringe connected to the irrigation channel of the sialendoscope through a connecting tube. The information found in the literature indicates that the intraductal pressures achieved with a 20 mL syringe are greater than or equal to those obtained with a 10 mL syringe, but in the former case, the second operator applies a much higher force when working with a larger syringe, without direct control of the pressure inside the salivary duct system [16]. While it is widely used, this procedure can be laborious, and it is often necessary to interrupt irrigation during surgery because of the high resistance to saline solution. Additionally, the pressure achievable by manual irrigation may not be sufficient to ensure an effective and permanent dilation of the stenoses. 

This paper presents a safe, less laborious, and simple method to perform efficient dilation of the salivary glands ducts during sialendoscopy using an adapted Cook Balloon Inflation Device. The patients were fully informed during the consent procedure of the risks associated regarding the off-label use of this medical device, and their informed consents were recorded. This revised irrigation system allows us to reach the desired pressure value and maintain it for as long as necessary, permitting optimal visualization of the duct and an effective progression of the endoscope into the ductal system. The saline pressure provides a radial force equally distributed in all directions in the ductal system and allows for the dilation of an entire stenotic tract, reaching a theoretical maximum pressure of 15 atmospheres on the ductal walls. The use of this device could allow the assisting nurse to completely replace the second operator in maneuvering the device during sialendoscopy.

During our clinical experience, no complications resulting from the use of this device were observed. The device is currently used without contraindications for all the patients who undergo sialendoscopic procedure.

## 4. Conclusions

The irrigation system described in this paper has proven to be more effective than a typical, often laborious, manual irrigation and can be widely used to perform dilation of the ducts of salivary glands during diagnostic or interventional sialendoscopy.

During sialendoscopy, a continuous rinsing of the duct is necessary to achieve dilatation of the duct and an adequate visualization of the lumen.Manual irrigation with medical syringe is sometimes tiring and unwarranted.The adapted balloon inflation device permits a non-laborious duct irrigation and dilation under pression control in any patient who undergoes sialendoscopy.

## Figures and Tables

**Figure 1 ijerph-19-14830-f001:**
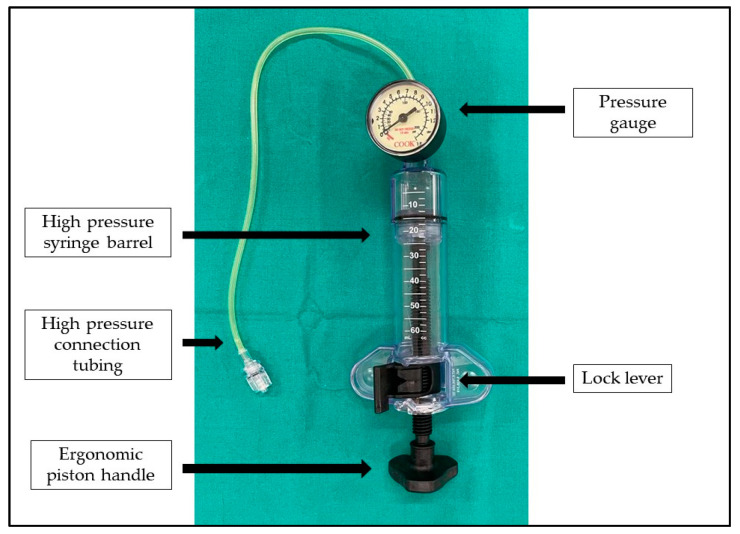
The irrigation device consists of a high-pressure syringe barrel, a lock lever, an ergonomic piston handle, a gauge used to monitor the inflation and deflation of balloon catheters, and connection tubing.

**Figure 2 ijerph-19-14830-f002:**
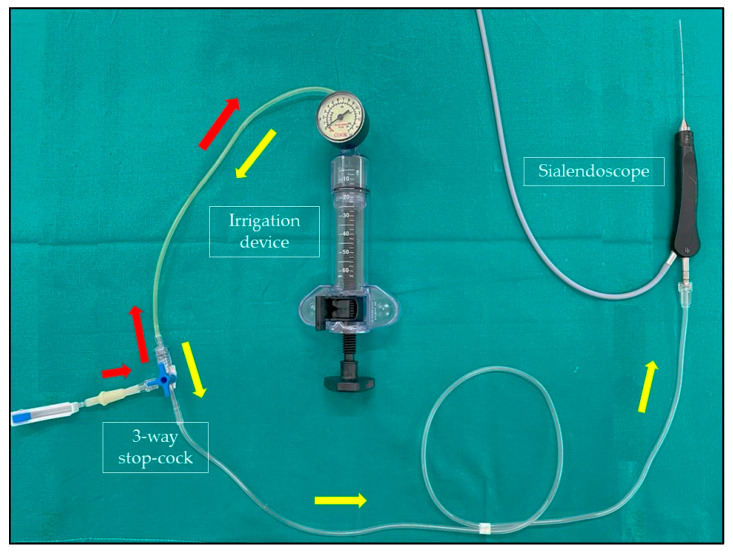
The irrigation device is connected with the irrigation channel of the sialendoscope and a saline solution bottle through a three-way stop-cock. Red arrows: saline solution is drawn into the syringe through the three-way stop-cock. Yellow arrows: saline solution is pushed into the sialendoscope through the three-way stop-cock under pressure control to perform an adequate duct dilatation.

**Figure 3 ijerph-19-14830-f003:**
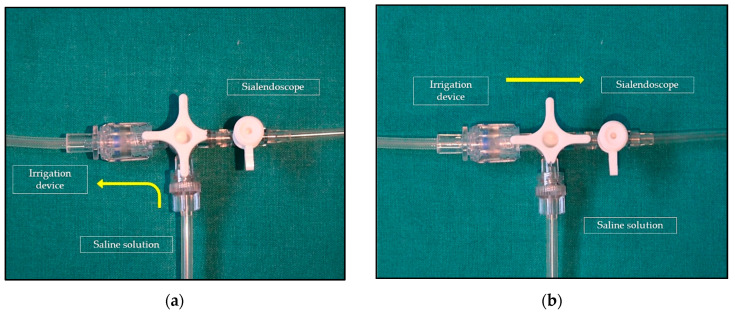
(**a**) The three-way stop-cock is positioned so as to allow saline to flow from the bottle to the syringe of the device, which can be filled with 20 cc of solution. (**b**) Once the syringe is filled with saline, the device is locked and the three-way stop-cock is rotated to allow fluid to pass from the syringe to the sialendoscope (**b**). The yellow arrows indicate the direction of the saline solution.

## Data Availability

Not applicable.

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
