# Peer review of "Use of Irrigation Device for Duct Dilatation during Sialendoscopy"

_ijerph, 2022, doi:10.3390/ijerph192214830_

Round 1
Reviewer 1 Report
Dear authors,
I have read your manuscrtipt with great interest. Since minimally invasive salivary gland surgery is evolving, it is important to address the methodological challenges of this ENT subspecialty and present them thoroughly. We have a vast experience of sialendoscopy in our centre; therefore, we share unpleasant experiences of salivary duct irrigation with classical syringes. However, the manuscript should be improved to be accepted for publication. Please see my comments below:
1.) ABSTRACT: please restructure after revising other parts of the manuscript
2.) HOW WE DO IT:
-It would be more informative and attractive to include a case presentation of your typical patient in this section.
-paragraph 1: mention other balloon dilation procedures in ENT in addition to FESS (Eustachian tube dilation, subglottic stenosis balloon dilation)
-line 48: please make sure that your procedure is not misunderstood as balloon sialoplasty.
-describe materials used in your manuscript in detail to be reproducible (manufacturer, model, reference number, country).
-Figures: describe what arrows mean.
3.) DISCUSSION
Discuss indications and contraindications, advantages, disadvantages or limitations of your procedure. All info of materials should not be described in discussion but in materials section (or how we do it).
-line 104: Place In this report in new paragraph
Author contributions: not needed, because this is not a research article.
Reviewer 2 Report
The article deal with an interesting topic and appears to be well written and scientifically sound, some minor issue could be improved.
Minor spell check is required
1) Numbering of the main of the main paragraphs is off. Please revise.
Discussion section:
2) line 82: Although it is true that salivary obstruction is the main indications several other inflammatory pathologies may benefit from this treatment. It could be interesting for the readers to compare the finding of this paper with
Colella G, Lo Giudice G, De Luca R, Troiano A, Lo Faro C, Santillo V, et al. Interventional sialendoscopy in parotidomegaly related to eating disorders. J Eating Disord 2021;9(1).
and
Lo Giudice G, Marra PM, Colella C, Itro A, Tartaro G, Colella G. Salivary Gland Disorders in Pediatric Patients: A 20 Years’ Experience. Appl Sci 2022;12(4).
3) line 111-116:
“The saline pressure provides a radial force equally in all directions of the ductal system and permits to dilate an entire stenotic tract reaching, theoretically, a maximum pressure value of 20 atmospheres on the ductal walls.”“When sialendoscopy is performed under local anesthesia, however, a pressure value exceeding two atmospheres is generally considered too painful for the patients. If it is necessary to reach higher pressure value, a general anesthesia could be required.”
Please provide reference for the athmospheres values comments.
6)Line 105 you define this device application as “cost effective”. Please try to compare the costs of the device to the regular syringe.
Is the assistant help avoidable at all?
Considering the nature of the paper it could be interesting a more deep comparison between this novel technique and the finding of the literature to better describe the differences and the improvement.
Moreover, the strong point of this technique could be synthesized into a bullet point paragraph in the conclusion
Round 2
Reviewer 1 Report
The authors improved the manuscript according to the review. Please revise:
-line 47: write the indications for Eustachian tube dilation, which is missing.
-line 90: ...inflammatory benign pathology. What about sialolithiasis without inflammation? Sialolithiasis is not considered an inflammatory disease unless there are signs of inflammation. There are some controversies in this sentence. Please clarify.
-Please clarify in line 110, "equal higher". Do you mean equal or higher? Please revise English.
I suggest the manuscript be accepted after this minor revision and English editing.
